# A new tool for evaluating health equity in academic journals; the Diversity Factor

Jack Gallifant[1]*, Joe Zhang[2], Stephen Whebell[3], Justin Quion[4], Braiam Escobar[5], Judy Gichoya[6], Karen Herrera[7], Ruxana Jina[8,9], Swathikan Chidambaram[10], Abha Mehndiratta[11], Richard Kimera[12,13], Alvin Marcelo[14], Portia Grace Fernandez-Marcelo[15], Juan Sebastian Osorio[16], Cleva Villanueva[17], Lama Nazer[18], Irene Dankwa-Mullan[19,20], Leo Anthony Celi[21,22,23]

1 Department of Intensive Care, Imperial College London NHS Trust, London, United Kingdom, 2 Institute of Global Health Innovation, Imperial College London, London, United Kingdom, 3 Intensive Care Unit, Townsville University Hospital, Townsville, Queensland, Australia, 4 University of the East Ramon Magsaysay Memorial Medical Center, Quezon City, Philippines, 5 Escuela de Ingeniería de Antioquia, Envigado, Colombia, 6 School of Medicine, Emory University, Atlanta, Georgia, United States of America, 7 Faculty of Medicine, Military Hospital, Managua, Nicaragua, 8 The Epidemiology and Surveillance Section, National Institute for Occupational Health, National Health Laboratory Services, Gauteng Region, South Africa, 9 The Wits School of Public Health, Faculty of Health Sciences, University of the Witwatersrand, Johannesburg, South Africa, 10 Department of Surgery and Cancer, Imperial College London, London, United Kingdom, 11 Center for Global Development, Washington, DC, United States of America, 12 Department of Information Technology, Faculty of Computing and Informatics, Mbarara University of Science and Technology, Mbarara, Uganda, 13 Department of Advanced Convergence, Handong Global University, Pohang-si, South Korea, 14 University of the Philippines Manila, Manila, Philippines, 15 Department of Family and Community Medicine, College of Medicine, University of the Philippines Manila, Manila, Philippines, 16 ScienteLab, Bogota, Colombia, 17 Instituto Politecnico Nacional, Escuela Superior de Medicina, Mexico City, Mexico, 18 Department of Pharmacy, King Hussein Cancer Center, Amman, Jordan, 19 Merative, & Center for AI, Research, and Evaluation, IBM Watson Health, Cambridge, Massachusetts, United States of America, 20 Department of Health Policy and Management, Milken Institute School of Public Health, George Washington University, Washington, DC, United States of America, 21 Massachusetts Institute of Technology, Laboratory for Computational Physiology, Cambridge, Massachusetts, United States of America, 22 Beth Israel Deaconess Medical Center, Division of Pulmonary, Critical Care, and Sleep Medicine, Boston, Massachusetts, United States of America, 23 Department of Biostatistics, Harvard T.H. Chan School of Public Health, Boston, Massachusetts, United States of America

☯ These authors contributed equally to this work.
* jack.gallifant@nhs.net

**Data Availability Statement:** All data have been sourced from open-access sources as described in the methods. These are freely available from the referenced sources without special privileges.

## Abstract

Current methods to evaluate a journal's impact rely on the downstream citation mapping used to generate the Impact Factor. This approach is a fragile metric prone to being skewed by outlier values and does not speak to a researcher's contribution to furthering health outcomes for all populations. Therefore, we propose the implementation of a Diversity Factor to fulfill this need and supplement the current metrics. It is composed of four key elements: dataset properties, author country, author gender and departmental affiliation. Due to the significance of each individual element, they should be assessed independently of each other as opposed to being combined into a simplified score to be optimized. Herein, we discuss the necessity of such metrics, provide a framework to build upon, evaluate the current landscape through the lens of each key element and publish the findings on a freely available website that enables further evaluation. The OpenAlex database was used to extract the metadata of all papers published from 2000 until August 2022, and Natural language

**Funding:** The authors received no specific funding for this work.

**Competing interests:** The authors have declared that no competing interests exist.

processing was used to identify individual elements. Features were then displayed individually on a static dashboard developed using TableauPublic, which is available at www.equitablescience.com/. In total, 130,721 papers were identified from 7,462 journals where significant underrepresentation of LMIC and Female authors was demonstrated. These findings are pervasive and show no positive correlation with the Journal's Impact Factor. The systematic collection of the Diversity Factor concept would allow for more detailed analysis, highlight gaps in knowledge, and reflect confidence in the translation of related research. Conversion of this metric to an active pipeline would account for the fact that how we define those most at risk will change over time and quantify responses to particular initiatives. Therefore, continuous measurement of outcomes across groups and those investigating those outcomes will never lose importance. Moving forward, we encourage further revision and improvement by diverse author groups in order to better refine this concept.

## Background

The last decade has seen our capacity to store, analyse and distribute health data grow exponentially, especially with the growing use of artificial intelligence (AI), yet healthcare has tried and failed to implement it in a successful manner. The current AI landscape is ever-expanding and many of the current models are either still in the prototype stage [1, 2] or exhibit substandard performance, particularly in the cases of sepsis and COVID-19 [3, 4]. More worryingly, AI has inherent bias, introduced by both the data and those who created it, and it is no surprise that it can disproportionately affect minorities [5, 6]. This has led to the call for greater transparency in the model development phase, improved data sharing, and more diversity among research groups to safeguard against such biases [7]. However, these changes have not yet reached the journal-level as the current metrics used to evaluate research and journal impact do not focus on such factors. Furthermore, the ability to provide a complete measure of health research's significance, penetrance, and relevance has been debated for decades [8].

Initially designed to track citations of articles by authors and journals, the impact factor (IF) is now used to judge the importance of scientific or academic publications and the journal itself. Though the IF accounts for variations in publishing volumes between journals, the impact on the population or a community had little relevance to the indices [9–12]. Currently, the IF has transformed into a proxy for the quality of individual articles even though highly cited papers skew calculations. As a result, journal IF figures do not represent the majority of papers published within a specific journal [13–16]. In addition, the IF has several limitations, such as not accounting for the citation density of fields or fluctuations in publication practices, for example, during the COVID-19 pandemic, which inflated critical care journals' impact factors [17, 18]. Moreover, leading academic journals have gamed the system to improve their own IF through self-citation, a practice which is equally common among leading authors [19, 20].

Despite the emphasis on IF, and citations as proxy, they do not equate to scientific excellence or the advancement of health research that improves outcomes for all [21]. There is increasing evidence of disparities in outcomes across demographic groups, where the COVID-19 pandemic was a prime example [22, 23]. These disparities are further reinforced by nonrepresentative research, in regards to the lack of diversity in both researchers and questions [24]. Having a research group that better represents their population in question allows for better coverage of multiple problem-solving styles and a better understanding of the problems they

face [25, 26]. Furthermore, increased geographical diversity of authorship has been found to be strongly correlated with scientific impact and can help lead to better science [27].

Yet, despite the heavy emphasis on the importance of diversity, there is no objective measure for tracking progress towards inclusivity in science or for evaluating who contributes to health research [7, 28]. A shift from a single citation-based metric to using several different metrics that provide a more complete perspective on factors aligned with scientific excellence based on contribution to advancing diversity and inclusion, improving health outcomes, and achieving equity is therefore necessary.

## Characteristics of a diversity factor for assessing journal impact

This proposed Diversity Factor (DF) should provide an alternative means of tracking accurate and reliable contributions to health research aligned with the impact on the global community or population in addition to offering an approach to facilitate scientific excellence that is unbiased, representative, and impactful. Important factors that should be considered in evaluating the literature include features related to publications, authorship, and research oversight. These are explored in Table 1 along with guiding questions that describe the feature's respective characteristics. It is important to note that this is merely a proposal and should serve as a foundation to build upon. Additionally, this proposed Diversity Factor should serve as a supplement to the currently used metrics and not as a replacement. Aggregation, scoring and weighting of each of these features requires rigorous survey multiple diverse stakeholder groups and is planned for future revisions. For this concept paper, each key element is reviewed independent of the others.

**1. Dataset characteristics.** Data selection is an inevitable component of research; not all data can be captured and instead strict selection is necessary to answer the research question at hand. However, this inherently creates a restricted view that affects the conclusions drawn, irrespective of domain. The properties of datasets used to develop medical devices and inform clinical decisions are vital as these conclusions will likely have the greatest relevance to the

**Table 1. Elements for assessing journal contribution to scientific excellence in diversity, equity and inclusion.**

| Category | Element | Guiding Questions |
|---|---|---|
| **Publications** | Diversity | Do studies explore health, determinants of health and underlying factors driving improved outcomes for diverse population groups? |
| | Equity | Do publications showcase research relevant to under-represented communities and populations? |
| | Inclusion | Have publications promoted inclusive and multidisciplinary research methods? |
| | Datasets | What are the definitions and distributions of age, race, ethnicity, gender, language, and geography, included in the study participants? |
| **Authorship** | Author identity, country or origin | Is there a wide range of author cultures, experience and expertise, including country or origin and low-middle income countries (LMICs)? |
| | Author Gender | Is there a balanced gender distribution on average among authors, and first/last authors? |
| | Author Organization and Affiliation | Is there a diverse range of organizations, including minority serving institutions, industry and academia? |
| | Community experts | Has the published research been conducted with community experts, and are they named co-authors? |
| **Research Oversight** | Journal Editors and Reviewers | Has the journal developed an accountability system to measure and ensure diversity among editors and reviewers? |
| | Review Process | Does a journal have a system to identify and respond to potential bias in the review process? |
| | Communication and Dissemination | Are the results of the above questions easily accessible and transparent for researchers and the general public to review? |

In this paper, the authors focused on developing tools that allow for the evaluation of the current landscape in regard to four key elements: dataset properties, author country of affiliation (LMIC status), author gender (M/F) and organisational affiliation (including multidisciplinary team approach).

populations studied [29]. The authors propose that datasets used in health research should be mapped to highlight geographic areas of data poverty, expose underlying knowledge gaps, and draw attention to imbalanced datasets [30]. Key properties of datasets that should be monitored include gender imbalance, race-ethnicity, language, age, and geography. The current impact factor rewards citations equally but, determining who has read, utilized, and has been 'impacted' by the research is not as simple as implied. Most data being analyzed to guide healthcare is derived from a few centers, almost exclusively based in High-Income Countries [31]. As such, increasing the global impact of research is vital, and the increasing adoption of technology has created the potential to democratize health research. More effort should then be placed on increasing the diversification of the data pool used to design clinical guidelines and develop tools that provide beneficial outcomes for all, and not just a select few countries.

**2. Author country.** Author's previous experiences, surrounding culture, and the associated team will significantly shape projects. Thus, when considering how to evaluate an author group, it is important to consider the diversity, as this will provide insight into what perspectives were considered in questions asked and conclusions reached. Evaluating the spread in the country of affiliations within studies can speak to the cognitive diversity of the teams and the likelihood of methodology and results transferring to that area. The authors designed the study methodology, conducted the analysis, interpreted the findings and presented these in an organized manner. Throughout these stages, biases can be introduced by influencing selection strategies, modes of analyses and presentation of results. Authors working in one country who analyze datasets from another have inherent limitations due to an incomplete understanding of the context and culture of the studied subjects. Including diverse perspectives can maximize the scope for identifying potential biases and ensuring the results produced are applicable to multiple populations. Notably, historical racial prejudices have resulted in disparities in clinical outcomes between demographics and including a variety of backgrounds would improve safeguarding against introducing similar biases [32]. It is commonplace for knowledge to centralize, with intellectual centers producing multitudes of research. These 'Ivory Towers' often overrepresent a particular demographic that is inconsistent with the experience or backgrounds of those most burdened by disease. Increasing diversity within author groups, especially within institutions, can help combat this resulting homogeneity of thought.

**3. Author gender.** Diversity is more than just increasing the number of ethnicities within the author group. Traditionally, academia has been a male-dominated field, well-documented across multiple fields [33, 34]. However within recent years, this trend has been shifting— more and more women are joining the field. However, gender-parity has not yet been reached and given the current trends will take several more years unless there are active, intentional changes. In this study, an algorithm trained to identify 'gender' as Male or Female was utilized. However, as further research is done to refine this proposal, expanding this definition to include other genders and the distinction between sex and gender would be ideal.

**4. Organizational or departmental affiliation.** Most health research has traditionally been conducted in a few institutions with the necessary funding and access to data. Considering the centralization of knowledge and the overrepresentation of certain demographics found in these institutions, there has been a recent shift to increase engagement with local stakeholders and wider population engagement. Further, healthcare is increasingly becoming a multidisciplinary field as a result of the recognition that socioeconomic factors play significant roles in health outcomes. As such, multidisciplinary teams play an important role in bridging professional boundaries and breaking down the barriers of competing cultural and organizational differences, thus rooting academic work in implementable applications. There are currently divides between clinical, academic, and commercial research that often leaves everyone feeling that data is out of reach. Understanding the current breakdown of institutional affiliation,

whether it is academic or commercial, is necessary to see if academia is responding to this multidisciplinary call. Expansion of this definition to include the composition of expertise will also be vital, for example, the interaction of machine learning engineers and social scientists in the field of AI.

### Analysis of diversity factor in journals with global reach

To evaluate the current diversity factor landscape, we used the OpenAlex database consisting of the metadata of all papers published from 2000 until August 2022 [35]. The entire database was downloaded locally, where metadata, including author name, affiliation, and study abstract, were extracted. Only journals identified by SCImago were included in the analysis.

Dataset Characteristics were not widely named in study abstracts, and due to the lack of freely available full papers, we could not identify this factor reproducibly. The code to implement this feature evaluation has been validated and is freely available for future use [36]. However, due to these limitations, it has been excluded from the analysis portion of this study.

Author affiliations were identified and geocoded using OpenAlex identification of more than 100,000 research producers in the Research Organisation Registry [37]. We geocoded raw affiliation strings for affiliations with no match using a custom Nominatim API [38]. Geographic locations were matched in 88% of author instances. Thereafter, Author Countries were grouped by income status according to the World Bank; Low-Middle Income Country (LMIC) or HIC Country [39].

Enriched metadata is produced using fine-tuned Natural Language Processing models (BERT-Pubmed) for research classification and entity extraction, as described elsewhere [1]. Affiliation strings derived from this process were then parsed for information on departmental affiliation that were then categorized into commercial and academic organizations.

We identified the author's gender using several APIs that demonstrate state-of-the-art performance in validation studies on non-English names, including Gender-API and Genderize [40, 41]. Gender matching was conducted using the first name and affiliation country, with 84% of entries matched. The female: male author ratio was calculated for each paper, and then a mean was calculated for each journal and time period.

Features are displayed individually, and a hypothesis of aggregation is discussed below. Descriptive statistics for each of the three included features is displayed in a subset of eight journals that cover the broadest ranges of speciality, Impact Factor, and traditional prestige. A static dashboard using over 7539 journals was then developed using TableauPublic to represent the full diversity factor landscape and to allow for effect modifiers (open-access and funding sources) to be evaluated; this is available at www.equitablescience.com.

### Results

Since the year 2000, 130,721 papers have been identified from 7539 journals, a majority of which are from authors based in North America or Central Asia and Europe. In 2021, there were 0.68 and 0.73 authors per paper per region, respectively, compared to under 0.1 in each of Latin America, the Caribbean, South Asia, and Africa. Underrepresentation of female authors is seen throughout. Taking the mean across all journals, in 2000, the median female: male author ratio per publication was 0.31 and has increased over time to 0.78 in 2021. There are a few countries where journals have reached gender parity, such as Portugal (1.13) and Cuba (1.09). However, despite this trend towards greater representation for female authors, it is unclear how many journals, if any, will reach gender-parity in the next five years given the rate of improvement. Those most at risk of being 'left behind' are primarily from low-income

countries though some HIC countries such as Japan (0.24) and South Korea (0.26) face such obstacles as well.

Authors from low-middle-income countries are sorely underrepresented as well. In 2021, there were over 5 million authors from high-income countries. In comparison, upper-middle-income countries had 1.5 million authors, lower-middle-income countries had 467,323, and low-income countries had 27,080 authors. Taking the mean across all journals, the median LMIC: non-LMIC author ratio per publication has increased from 0.04 in 2000 to 0.20 in 2021 (Fig 1B)

Interestingly, these trends of female author and LMIC author underrepresentation are reversed when looking at a subset of open-access journals. When comparing the top 25 open access journals and non-open access journals by IF, open access journals consistently had higher proportions of female authors and authors from LMICs over the past two decades.

Table 2 summarizes the findings from a selection of well-known journals from 2021. The female:male ratio of authors in this subset mirrors the consistent pattern of female author underrepresentation seen in all journals, with none from this subset averaging over 1. Author

A

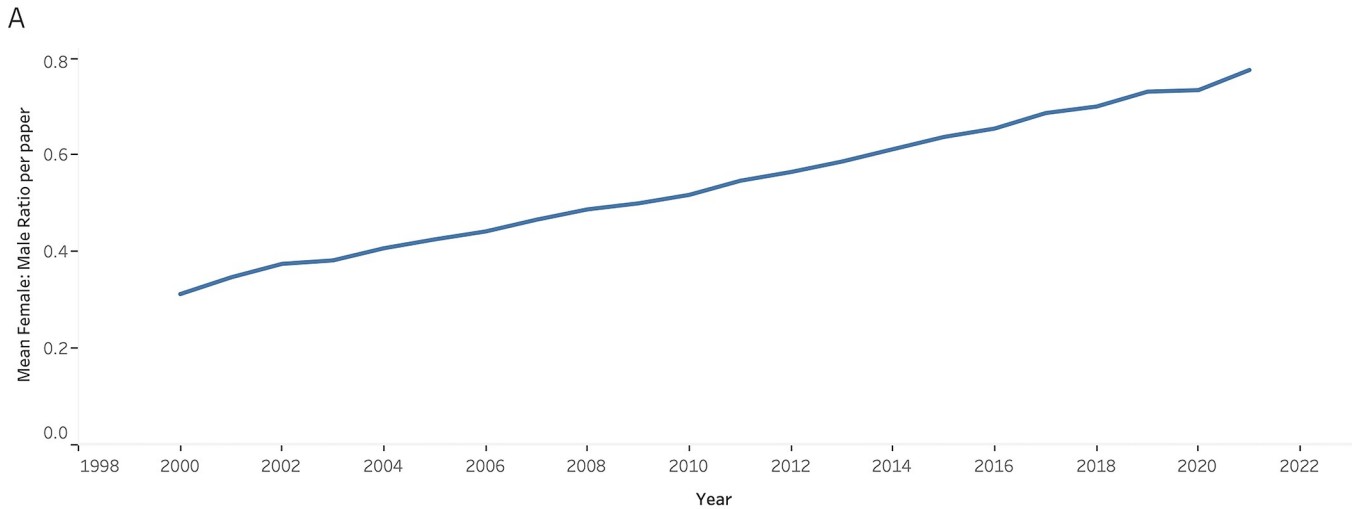

B

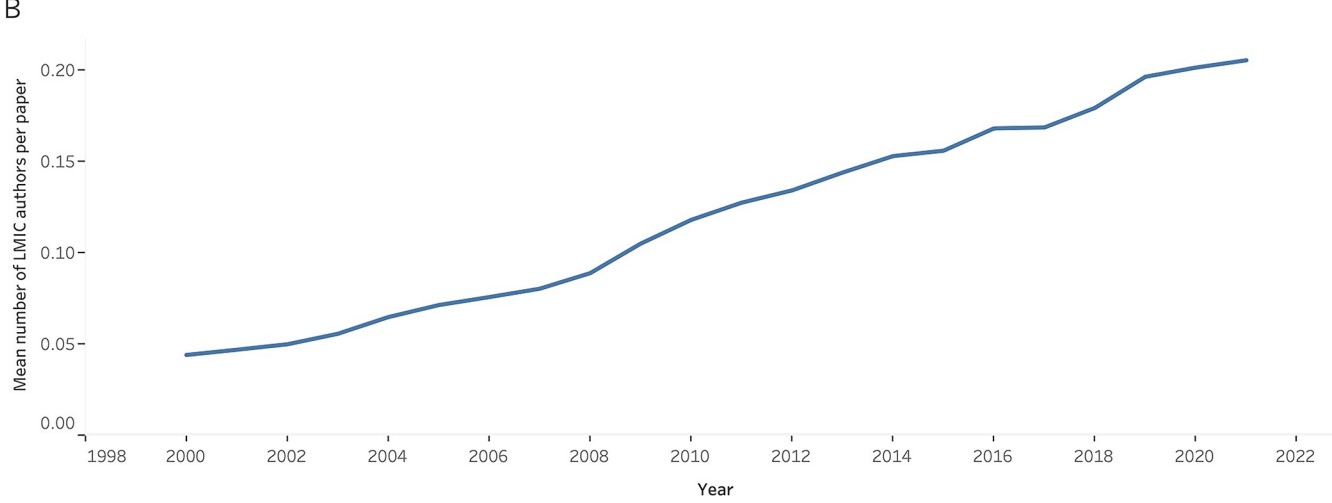

**Fig 1.** The mean value for **A)** median Female: Male author ratio on a paper per journal each year from 2000–2021, **B)** median LMIC: non-LMIC author ratio per journal each year from 2000–2021.

**Table 2. Comparison of median diversity factors across journals in 2021.**

| Journal | SJR (IF) | Female: Male * | LMIC: non-LMIC author * | Commercial: Non-commercial affil.* |
|---|---|---|---|---|
| BMJ | 2291 (93) | 0.43 | 0.03 | 0.02 |
| JAMA | 6076 (157) | 0.40 | 0.01 | 0.03 |
| Lancet | 6024 (202) | 0.37 | 0.15 | 0.03 |
| Nature Medicine | 15652 (87) | 0.38 | 0.02 | 0.05 |
| NEJM | 24161 (176) | 0.36 | 0.03 | 0.09 |
| PLoS Medicine | 24907 (12) | 0.42 | 0.12 | 0.01 |
| npj Digital Medicine | 4180 (15) | 0.30 | 0.02 | 0.12 |
| The Lancet. Digital health | 3326 (37) | 0.31 | 0.03 | 0.05 |

SJR = Scimago Journal Ranking, Average number of weighted citations received in a year by articles published in a journal in the year 2021. IF = Impact Factor.

* Ratio of mean values per paper for each journal 2dp. Commercial affil. = the mean number of authors per paper from a commercial company.

representation from LMICs is consistently low as well, with most journals under 1:10. In addition, most authors were from academic or non-commercial organizations, highlighting the lack of multidisciplinary collaboration.

## Discussion

In this paper, the concept of a Diversity Factor was proposed as a supplemental metric of measuring a journal's contribution to the research landscape, focusing on diversity, equity, inclusion, and impact on the studied community or population. Analysis of the data under the lens of the key elements of a proposed Diversity Factor reveal unsurprising results. Female authors and authors from LMICs are sorely underrepresented. While trends are improving for both demographics, several obstacles still stand in the way. Academia is becoming more and more centralized especially in high-income countries. As it does so, it becomes more difficult to penetrate, especially when considering the trends of self-citation.

The next steps for implementing a diversity factor would include the availability of dataset characteristics, detailed funding sources, patents and downstream policy impact, and citation mapping. This would allow for a better understanding of who is impacted and who is causing the impact. Additionally, refining the use of author country to a place of birth or time spent in a country would account for those in LMICs who emigrate to other institutions, which is not uncommon. Further, the improvement of gender data or NLP algorithms to account for author gender compared to sex would be another step forward. The operationalisation of this tool would rely on interest from journals and researchers in collaborating to permit this data to be published on each journal's website and centrally for evaluation. The heterogeneity in the findings between these four diversity metrics and between journals likely means that these four values should not be combined into one 'diversity value'. Instead, they should be evaluated and compared individually as seen in Fig 2, or other visual tools such as in a star diagram.

Additionally, we propose having such values rechecked yearly to offer checkpoints and track progress as new research is published and populations evolve. Routine reporting would further allow for more detailed analysis to be performed, highlight gaps in knowledge, and reflect confidence in the translation of related research. This is particularly true at the health policy level, where it is known that the social determinants of health vary greatly between countries; therefore, clinical decisions and public health decisions should be made based on information more representative of these populations. It is important to acknowledge that there is not one type of bias, nor one group affected by bias, but many types of bias and many groups that can be biased against. Diversity is not a box-ticking exercise but an essential

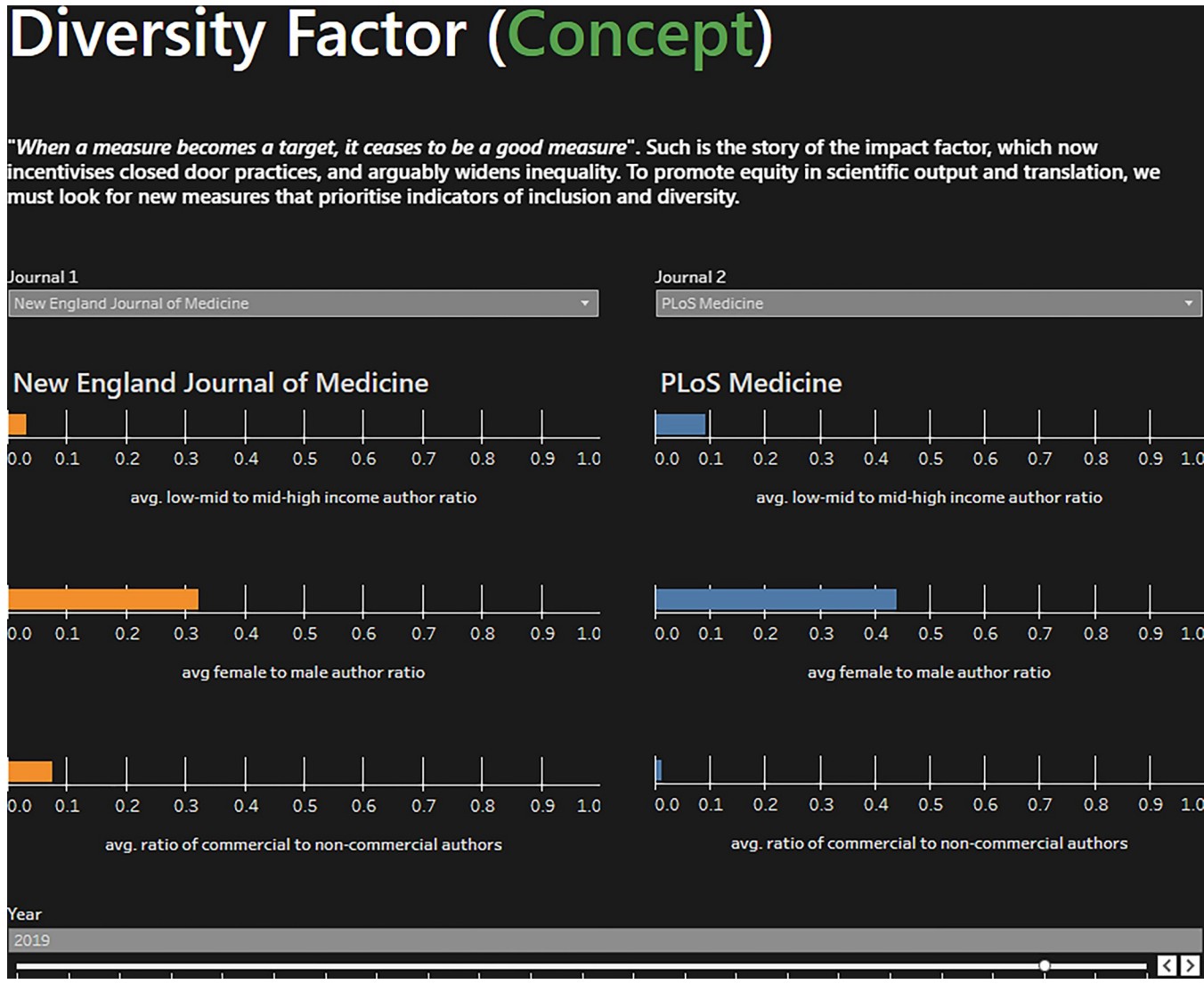

**Fig 2. Comparison of the individual features of the diversity factor for two journals, from https://equitablescience.com/dashboards/diversity/.**

safeguard against potential biases, especially in countries with greater ethnicity, cultural diversity and particular socio-demographic characteristics. Promotion of these features would encourage thoughtful discourse on how study designs and data characteristics can affect different groups so that researchers and organizations can build the right team for the specific project and its risks. Furthermore, integration of the Diversity Factor can encourage collaboration with LMICs that could reshape the knowledge landscape through the dissemination of work partnered with LMICs.

Our goal was to evaluate the current diversity factor landscape to the highest degree permissible by the availability of data and the current standards of metadata. Data set characteristics were not widely available, which did not permit the evaluation of this important feature. However, we have provided the tools to implement this feature if made available in the future. Gender was determined using NLP, which has demonstrated good performance across countries; however, the ground truth is not available for these studies and so cannot be confirmed. SCImago Journal Ranking is used in place of traditional impact factor due to the open source

nature and availability of the feature, there is a similarity between the two, but we also acknowledge differences. We recognise these tools are imperfect, but we hope they provide a 'big picture' view of the global research landscape and demonstrate what is possible in the future with greater open access to journals and why these metrics could greatly promote the drive for equitable science.

The four elements used in this paper uncovers a bleak reality unseen by citation-based metrics. Academia as it is now, and the healthcare systems that it shapes, cannot equitably and justly provide for all if it is not reflective of the populations in question. And in order to do that, it is necessary to change the way science evaluates its efforts. The current methods of measuring journal impact are far from ideal and fail to provide an estimate of author and dataset diversity. While diversity should not be the only consideration of researchers or journals, it should complement the downstream citation impact. Yet, tracking who participates in the conversations that shape healthcare and where our opinions are being formed should be monitored and evaluated transparently and publicly.

The Diversity Factor is a call to action for improved representation and the encouragement of diverse perspectives in health research to prevent the perpetuation of biases against subgroups and the advancement of scientific excellence that works for all. It reminds journals and authors to assess how thoughts and data reach the manuscript and whether they consider all perspectives, not just those available at hand. Otherwise, we will continue learning and practising medicine in an echo chamber created by the few ivory tower academics with access to the resources and data required to advance the field left in the hands of a select few institutions.

These findings and a more detailed analysis from this paper are available online to permit comparisons across open access, funding sources and specific use cases such as COVID-19 [42]. Moreover, the code is also freely available online in the interest of reproducibility, the addition of other features, and for later conversion to an active pipeline [36].

## Acknowledgments

We would like to acknowledge the contribution of the OpenAlex database, without which this project would not have been possible.

## Author Contributions

**Conceptualization:** Jack Gallifant, Joe Zhang, Richard Kimera, Lama Nazer, Irene Dankwa-Mullan, Leo Anthony Celi.

**Data curation:** Jack Gallifant, Joe Zhang, Stephen Whebell.

**Formal analysis:** Jack Gallifant, Joe Zhang, Stephen Whebell, Justin Quion.

**Investigation:** Jack Gallifant, Joe Zhang.

**Methodology:** Jack Gallifant, Joe Zhang, Stephen Whebell, Leo Anthony Celi.

**Project administration:** Jack Gallifant, Joe Zhang, Leo Anthony Celi.

**Software:** Stephen Whebell.

**Supervision:** Irene Dankwa-Mullan, Leo Anthony Celi.

**Validation:** Jack Gallifant, Joe Zhang, Stephen Whebell, Justin Quion, Leo Anthony Celi.

**Visualization:** Jack Gallifant, Joe Zhang, Stephen Whebell, Irene Dankwa-Mullan, Leo Anthony Celi.

**Writing – original draft:** Jack Gallifant, Joe Zhang.

**Writing – review & editing:** Jack Gallifant, Joe Zhang, Stephen Whebell, Justin Quion, Braiam Escobar, Judy Gichoya, Karen Herrera, Ruxana Jina, Swathikan Chidambaram, Abha Mehndiratta, Richard Kimera, Alvin Marcelo, Portia Grace Fernandez-Marcelo, Juan Sebastian Osorio, Cleva Villanueva, Lama Nazer, Irene Dankwa-Mullan, Leo Anthony Celi.

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
