## [Decision Letter · Decision Letter 0]

5 Apr 2023

PGPH-D-23-00134

A new tool for evaluating health equity in academic journals; the Diversity Factor

Dear Dr. Gallifant,

Thank you for submitting your manuscript to PLOS Global Public Health. After careful consideration, we feel that it has merit but does not fully meet PLOS Global Public Health’s publication criteria as it currently stands. Therefore, we invite you to submit a revised version of the manuscript that addresses the points raised during the review process.

We look forward to receiving your revised manuscript.

Kind regards,

Zahra Zeinali, MD MPH DrGH (c)

Academic Editor

Journal Requirements:

1. Please note that PLOS has specific guidelines on code sharing for submissions in which author-generated code underpins the findings in the manuscript. In these cases, all author-generated code must be made available without restrictions upon publication of the work. Please review our guidelines at https://journals.plos.org/globalpublichealth/s/materials-and-software-sharing#loc-sharing-code and ensure that your code is shared in a way that follows best practice and facilitates reproducibility and reuse.

2. Please amend your online Financial Disclosure statement. If you did not receive any funding for this study, please simply state: “The authors received no specific funding for this work.”

3. Please update your online Competing Interests statement. If you have no competing interests to declare, please state: “The authors have declared that no competing interests exist.”

4. Please provide separate main figure files in .tif or .eps format only and remove any figures embedded in your manuscript file. Please also ensure that all files are under our size limit of 10MB.

Additional Editor Comments:

Thank you for submitting your manuscript, entitled "A new tool for evaluating health equity in academic journals; the Diversity Factor," to PLoS GPH. We appreciate the effort and time you have invested in conducting your research and preparing your manuscript for our consideration.

Our peer reviewers have thoroughly assessed your submission, and I regret to inform you that your manuscript requires major revisions before it can be reconsidered for publication in our journal. The primary reasons for this decision are as follows:

[Issue 1]: The manuscript requires an overall strengthening of the argument. For example, concepts such as commercial and non-commercial fields, ITA, etc., can be better explained and expanded on.

[Issue 2]: The literature review needs to be expanded to include more recent and relevant studies and a more comprehensive analysis of the existing body of research.

[Issue 3]: The connection between authors' diversity and research impact needs to be clearly defined. The manuscript would strongly benefit from a more robust and nuanced explanation of this connection.

[Issue 4]: The connection of the diversity factor and the field of digital health needs to be described in more detail as it currently leads to more questions than answers. Please review this by either highlighting this connection or by avoiding using the term digital health, as it entails a specific definition of a particular field.

[Issue 5]: Line 158, cognitive diversity does not necessarily convey the diversity of expertise, lived experiences, or points of view and I encourage you to revisit this phrase/concept.

[Issue 6]: Line 175, 176, there is an underlying assumption that having female authors (better to use gender terms instead of sex terms, i.e., women authors) would automatically reduce the bias and safeguard against inequitable practices. However, this does not necessarily hold as both men and women operate within a patriarchal culture and system and one's sex and/or gender does not automatically create immunity against and departure from patriarchal practices.

[Issue 7]: Line 231, 232 reads "Additionally, in 2021, compared to on average."

[Issue 8]: Lines 244 to 246, it is unclear what kind of collaboration would improve the impact, if that applies to all types of research mentioned in this manuscript, how collaboration is defined and implemented. This concept of collaboration can benefit from further exploration and elaboration.

[Issue 9]: Using phrases such as "prestigious subset" in lines 245, 246 creates a tone that does not necessarily go along with the transformative ideas behind the manuscript.

Please find the detailed reviewer comments and suggestions in the attached file. We strongly encourage you to carefully address each of the issues raised by our reviewers and provide a point-by-point response to their feedback in your revised submission.

We understand that major revisions require significant time and effort, but we believe that addressing these concerns will greatly enhance the quality, rigor, and impact of your manuscript. If you decide to revise your manuscript, please submit the revised version along with your point-by-point response within six weeks.

Once again, thank you for considering PLoS GPH as a venue for your work. We look forward to receiving your revised manuscript, and please do not hesitate to contact us if you have any questions or need further clarification.

Sincerely,

Dr. Z. Zeinali

Academic Editor

Reviewers' comments:

Reviewer's Responses to Questions

**Comments to the Author**

1. Does this manuscript meet PLOS Global Public Health’s publication criteria? Is the manuscript technically sound, and do the data support the conclusions? The manuscript must describe methodologically and ethically rigorous research with conclusions that are appropriately drawn based on the data presented.

Reviewer #1: Yes

2. Has the statistical analysis been performed appropriately and rigorously?

Reviewer #1: Yes

3. Have the authors made all data underlying the findings in their manuscript fully available (please refer to the Data Availability Statement at the start of the manuscript PDF file)?

Reviewer #1: Yes

4. Is the manuscript presented in an intelligible fashion and written in standard English?

Reviewer #1: Yes

5. Review Comments to the Author

Reviewer #1: Abstract:

• This is mentioned in the abstract after mentioning the “diversity factor” – “Herein, we develop the methodology to reproducibly calculate these elements, evaluate the current landscape and publish the findings on a freely available website that enables further evaluation”. However, this sentence failed to establish consistency and its relationship with the diversity factor. Hence, some restructuring is advised

Overall:

• The authors should have described how the diversity factor would be scored which can help the readers to visualize and compare the factor between journals

• The article is well written. But yet, it will be hard to convince that this diversity factor would replace the current impact factor system of the journals. In my opinion, the authors can revise their hypothesis, and claim that this diversity factor should be included while measuring the impact factor of a journal.

6. PLOS authors have the option to publish the peer review history of their article (what does this mean?). If published, this will include your full peer review and any attached files.

**Do you want your identity to be public for this peer review?** For information about this choice, including consent withdrawal, please see our Privacy Policy.

Reviewer #1: No

---

## [Editor Report · Decision Letter 1]

25 Jun 2023

PGPH-D-23-00134R1

A new tool for evaluating health equity in academic journals; the Diversity Factor

Dear Dr. Gallifant,

Thank you for submitting your manuscript to PLOS Global Public Health. After careful consideration, we feel that it has merit but does not fully meet PLOS Global Public Health’s publication criteria as it currently stands. Therefore, we invite you to submit a revised version of the manuscript that addresses the points raised during the review process.

A rebuttal letter that responds to each point raised by the editor. You should upload this letter as a separate file labeled 'Response to Reviewers'.A marked-up copy of your manuscript that highlights changes made to the original version. You should upload this as a separate file labeled 'Revised Manuscript with Track Changes'.An unmarked version of your revised paper without tracked changes. You should upload this as a separate file labeled 'Manuscript'.

We look forward to receiving your revised manuscript.

Kind regards,

Zahra Zeinali, MD MPH DrGH (c)

Academic Editor

Journal Requirements:

Additional Editor Comments (if provided):

Thank you so much for sending us your revised manuscript. It has improved significantly.

There are a few minor edits that I'd like to suggest for you to consider before we proceed. Please review them and submit your revised version (v2) at your earliest convenience.

Thank you,

Dr. Zeinali

Editor comments:

Line 126: used to? does this not hold true anymore?

Line 132: journal IF figures

Line 145: as well as the research population

Line 148: ethnic? I think it would be more comprehensive if you mentioned something along the lines of diversity of authors, such as geographic diversity of where they are from and where they are currently working.

Line 155: inclusion?

Line 171: publication, authorship, reseach oversight - any reason why you have chosen to make the first letetrs capital?

Line 175: scoring of this… needs a bit more elaboration

Line 189: country as in country they live in or country they are based at?

Line 192: properties or characteristics?

Line 193: change inevitable with critical

Line 198: the point about AI is very out of place here. this is not a paper about AI.

Line 207: this is unnecessarily complex - geographic and academic areas of data poverty, what is exactly meant? And then what does these in line 207 refer to?

Line 215: see if the editorial of lancet globalhealth isseue June has relevance to this part

Lines 227-228: these are assumptions that need to be substantiated further

Line 237: locations or populations?

Line 252: again, ethnicities is too narrow a word and concept. You probably mean citizenship, nationality, race, backgrouns, etc. + ethnicity

Line 273: institutions, or ivory tower institutions

Lines 274-275: patient public involevement? sometimes it’s not only patients, but the general population under study. I recommend changing this to increase population engagement.

Line 280: will does not pertain to a specific subject - multidisciplinary teams’ verb is play

Line 281: medical devices have not been discussed before and don’t really seem relevant here. deploy or implement?

Line 284: repeated from 280 - 281

Line 294: were extracted (data and metadata are plural)

Line 308: any specific reason that you chose non-LMIC instead of HIC?

Line 332: which region? can you introduce all the regions considered?

Line 335: journals? or the peer-reviewed publications in journals?

Line 336: is the number the female to male ratio? you need to clarify this.

Line 339: some HIC such as …

Line 349: the 2000 ratio is worth mentioning here

Line 352: most impactful or with the highest IF?

Lines 393-396: this paragraph needs to go further down, and fully rewritten for clarity and ease of understanding.

Line 488: remove capital letters

Line 521: Table

1. Comment on guiding questions for publication diversity: publication or dataset? this needs to be consistent across the paper.

2. Comment on guiding questions for publication inclusion: what are inclusive research methods?
---

## [Editor Report · Decision Letter 2]

14 Jul 2023

A new tool for evaluating health equity in academic journals; the Diversity Factor

PGPH-D-23-00134R2

Dear Dr. Gallifant,

We are pleased to inform you that your manuscript 'A new tool for evaluating health equity in academic journals; the Diversity Factor' has been provisionally accepted for publication in PLOS Global Public Health.

Best regards,

Zahra Zeinali, MD MPH DrGH (c)

Academic Editor